# A Set of New Stable, Explicit, Second Order Schemes for the Non-Stationary Heat Conduction Equation

**Endre Kovács *** 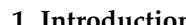, **Ádám Nagy** and **Mahmoud Saleh**

Institute of Physics and Electrical Engineering, University of Miskolc, 3515 Miskolc, Hungary; fizadam@uni-miskolc.hu (Á.N.); mhmodsalh84@gmail.com (M.S.)
* Correspondence: fizendre@uni-miskolc.hu

**Abstract:** This paper introduces a set of new fully explicit numerical algorithms to solve the spatially discretized heat or diffusion equation. After discretizing the space and the time variables according to conventional finite difference methods, these new methods do not approximate the time derivatives by finite differences, but use a combined two-stage constant-neighbour approximation to decouple the ordinary differential equations and solve them analytically. In the final expression for the new values of the variable, the time step size appears not in polynomial or rational, but in exponential form with negative coefficients, which can guarantee stability. The two-stage scheme contains a free parameter $p$ and we analytically prove that the convergence is second order in the time step size for all values of $p$ and the algorithm is unconditionally stable if $p$ is at least 0.5, not only for the linear heat equation, but for the nonlinear Fisher's equation as well. We compare the performance of the new methods with analytical and numerical solutions. The results suggest that the new algorithms can be significantly faster than the widely used explicit or implicit methods, particularly in the case of extremely large stiff systems.

**Keywords:** heat equation; explicit time-integration; stiff equations; parabolic partial differential equations; unconditional stability

## 1. Introduction

It is generally known that conductive heat transfer is described by a second-order linear parabolic partial differential equation (PDE), the so-called heat equation:

$$\frac{\partial u}{\partial t} = \alpha \nabla^2 u + q, \tag{1}$$

or more generally

$$c\rho \frac{\partial u}{\partial t} = \nabla(k\nabla u) + c\rho q \tag{2}$$

where $u = u(x,t)$ or, in the case of Equation (2), $u = u(\vec{r},t)$ is the temperature, $\alpha = k/(c\rho) > 0$ is the thermal diffusivity, $q$, $k$, $c$, and $\rho$ are the intensity of heat sources (electromagnetic radiation, chemical reactions, radioactive decay, etc.), heat conductivity, specific heat and (mass) density, respectively. These equations and their generalizations such as the convection (or advection)–diffusion equation and reaction–diffusion equation are used to model the diffusion of various particles in chemical and biological systems [1] as well as in electronic devices [2]. Mathematically very similar equations are used to treat the flow of fluids through porous media [3], like moisture [4], ground water or crude oil in reservoirs. When diffusion or fluid flow is modelled, the $u$ variable denotes the concentration or the pressure. We will examine one of the nonlinear reaction–diffusion equations, namely Fisher's equation [5], which originally was introduced to model how advantageous gene-variants spread in space and time.

As applied science and measurement techniques are developing, higher and higher resolutions are required in numerical simulations. Because of this trend, the investigated spatial domain has to be divided into smaller and smaller cells during the discretization process, which means that the number of cells, i.e., the number of unknowns and that of equations are also rapidly growing. On the other hand, in real systems, the physical properties like the specific heat and the heat conductivity can be completely different at different points close to one another ([6], p. 15). Therefore the coefficients and thus the eigenvalues of the problem may have a range of several orders of magnitude, which implies that the problem can be highly stiff. It is well known that conventional explicit methods (either Runge-Kutta or multistep Adams-Bashforth types) are only conditionally stable; thus, the time steps must be reduced to very small values regardless of the actual accuracy requirements. This is why the implicit methods with much better stability properties are widely considered to be superior [7,8] and almost exclusively used to solve these kinds of problems [9]. On the other hand, they require the solution of a system of algebraic equations at each time step, often by iterative methods. For huge matrices their solution is still rather slow, especially in more than one space dimension, where the matrix is not tridiagonal. To overcome the difficulties, enormous efforts have been made and numerous techniques have been developed. More and more sophisticated preconditioners are used [10,11], geometric, algebraic and hybrid (i.e., combined geometric and algebraic) multigrid systems are applied [12,13] and even the implicit Euler method is supplemented by a linear time filter to remove its overdamping and increase its accuracy [14]. These improvements are impressive, but unfortunately, the parallelization of implicit algorithms is still not straightforward, although some moderate progress has been achieved [15,16]. However, as the clock frequency increase of CPUs has drastically slowed down in the last two decades, the trend towards increasing parallelism continues in high performance computing [17].

Nevertheless, it is not true that all explicit methods have such restrictive stability conditions. To the contrary, a couple of explicit methods exist which are more stable or even unconditionally stable. The most notable examples are the Dufort-Frankel scheme ([18], p. 88), the Runge–Kutta–Chebyshev methods [19], the Hopscotch method [20,21], Alternating Direction Explicit (ADE) and Alternating Group Explicit (AGE) methods [22–24], the D'Yakonov fully explicit variant of the iterative alternating decomposition method [25] and the positivity preserving schemes of Chen–Charpentier and others [26,27]. We have to note that, according to our knowledge, these methods haven't even been listed together in any single document apart from our work, let alone being systematically tested against one another and against popular methods. These methods also have weak points, for example, some of them are only conditionally convergent or consistent, they are usually less accurate, they can be complicated to code or hardly be applied for irregular grids [28–32]. These disadvantages explain, at least partially, that the investigation and application of these methods are rather rare (see for example [4,16,33,34]) and they remained relatively unknown. Even those scholars who work with semi-implicit schemes, also called mixed explicit–implicit schemes [35], do not really use these explicit methods. From the information collected above, one may draw the conclusion that known methods do not provide a truly convenient solution for this problem. In order to fill this growing gap, we started to elaborate a class of fundamentally new explicit methods. The simplest, first order member of this class has been already published [36]. At the beginning of the next section, we restate this simple method, because this is the basis of the second order methods; thus, it is necessary to understand this first. Then we introduce the new algorithms, first for the simplest, one dimensional grid, then for general, arbitrary grids. In Section 3 we analyse their properties, then we turn to the numerical tests in the next sections. We present here two analytical cases in Section 4 for verification (heat equation and Fisher's equation) and then two numerical cases in Section 5. We mention that the method we introduce here can be considered as a generalization of our previous CN2 method [37], which was only first order because it used only integer time steps.

## 2. The Description of the Methods

To solve Equation (1) numerically, we begin with the most typical starting step, as in the standard method of lines. We discretize the second space derivatives by the most widely used second order central difference formula ([38], p. 48)

$$\frac{\partial^2}{\partial x^2} f(x_i, t_j) \approx \frac{\frac{f(x_{i+1}, t_j) - f(x_i, t_j)}{\Delta x} + \frac{f(x_{i-1}, t_j) - f(x_i, t_j)}{\Delta x}}{\Delta x} . \tag{3}$$

First, we examine the one dimensional heat equation in a homogeneous medium. After defining an equidistant grid of $N$ grid nodes we get the following ordinary differential equation (ODE) system:

$$\frac{du_i}{dt} = \alpha \frac{u_{i-1} - 2u_i + u_{i+1}}{\Delta x^2} + Q_i,$$

where $Q_i$ is the value of the $q(x)$ source function at node $i$. This system of equations can be written in a briefer matrix-form:

$$\frac{d\vec{u}}{dt} = M\vec{u} + \vec{Q} . \tag{4}$$

The diagonal elements of the matrix $M$ are the sums of the non-diagonal elements of the appropriate rows with a negative sign:

$$m_{ii} = -\frac{2\alpha}{\Delta x^2} \quad (1 < i < N), \quad m_{11} = m_{NN} = -\frac{\alpha}{\Delta x^2} .$$

We introduce the characteristic time or time-constant of the cell i:

$$\tau_i = \frac{-1}{m_{ii}} = \frac{\Delta x^2}{2\alpha} .$$

### 2.1. The One-Stage Constant-Neighbour (CN) Method

In this point we will recall the already published [36] method to solve the ODE system (4), which can be derived through the following steps:

1.  We make an approximation: When we calculate the new value of a variable $u_i^{n+1}$, we neglect that other variables are also changing during the time step $h = \Delta t$. This means that we consider $u_j$ a constant if $j \neq i$, so we have to solve uncoupled, linear ODEs:

$$\frac{du_i(t)}{dt} = a_i - \frac{u_i(t)}{\tau_i} \tag{5}$$

where the $u_i$ unknowns are considered as a function of time with initial values $u_i^n$, which are the temperatures at the beginning of the $n$-th time step. We introduced

$$a_i = \sum_{j \neq i} m_{ij} u_j^n + Q_i = \alpha \frac{u_{i-1}^n + u_{i+1}^n}{\Delta x^2} + Q_i .$$

2.  The ODE system (5) is a highly simplified approximation of the original PDE, and it is straightforward to solve it analytically:

$$u_i(t) = u_i^n \cdot e^{-\frac{t}{\tau_i}} + a_i \tau_i \cdot \left(1 - e^{-\frac{t}{\tau_i}}\right) \qquad 0 < t < h .$$

We use this solution at the end of the time step to obtain the new values of the $u$ variable:

$$u_i^{n+1} = u_i^n \cdot e^{-\frac{h}{\tau_i}} + a_i \tau_i \cdot \left(1 - e^{-\frac{h}{\tau_i}}\right) . \tag{6}$$

For a standard one dimensional homogeneous system with an equidistant grid, we have the following one-stage first order method, which was called "constant-neighbour method" or CN scheme.

The CN method algorithm

$$u_i^{n+1} = u_i^n \cdot e^{-\frac{2\alpha h}{\Delta x^2}} + \left[ \frac{u_{i-1}^n + u_{i+1}^n}{2} + \frac{\Delta x^2 Q_i}{2\alpha} \right] \left( 1 - e^{-\frac{2\alpha h}{\Delta x^2}} \right) \tag{7}$$

while the actual values can be denoted as $u_i^{CN}$. Formula (7) can be written as:

$$u_i^{n+1} = u_i^n + \left( \frac{u_{i-1}^n - 2u_i^n + u_{i+1}^n}{2} + \frac{\Delta x^2 Q_i}{2\alpha} \right) \left( 1 - e^{-\frac{2\alpha h}{\Delta x^2}} \right) \tag{8}$$

where the simple identity $e^\varphi = 1 + (e^\varphi - 1)$ has been applied. Using the power series of the exponential function, we get:

$$1 - e^{-\frac{2\alpha h}{\Delta x^2}} = \frac{2\alpha h}{\Delta x^2} - 2 \left( \frac{\alpha h}{\Delta x^2} \right)^2 + \dots .$$

Inserting this back to the previous (8) formula and keeping only the first term yields:

$$u_i^{n+1} = u_i^n + \alpha \frac{u_{i-1}^n - 2u_i^n + u_{i+1}^n}{\Delta x^2} h + Q_i h$$

which is exactly the explicit Euler method, also called Forward Time Central Space (FTCS) scheme in the case of PDEs. One can see that regarding one time step, the one-stage CN method is identical to the explicit Euler method, but only up to first order. However, in spite of this apparent similarity, there is a fundamental difference between the methods: During step (1) we do not fix the actual variable $u_i$ to its initial value (we fix only the neighbours), but keep it as a variable. This means that at the end of this step we still have differential equations and not difference equations to be solved analytically, i.e., here time derivatives are not approximated by finite difference formulas as in Finite Difference Methods (FDM). In our CN method, the time step size $h$ in the Euler/FTCS scheme is replaced by $\left[ 1 - \exp\left( -\frac{2\alpha h}{\Delta x^2} \right) \right] \frac{\Delta x^2}{2\alpha}$ and because of these exponential terms, Formulas (6)–(8) contain $h$ up to infinite order, which is crucial for the unconditional stability.

### 2.2. The Two-Stage Constant-Neighbour Method

Now we show how to create a more accurate, second order two-stage "combined CN" algorithm. In Table 1, one can see the Butcher tableau of the well-known explicit generic second order Runge-Kutta (RK) method. In three cases these methods obtained distinct names:

- for $p = 1/2$, it is called the midpoint method.
- for $p = 1$, one has the explicit trapezoidal rule, which is also called Heun's method or improved Euler's method.
- for for $p = 2/3$, it is called the Ralston-method, which provides a minimum bound on the truncation error for the second-order RK algorithms [39].

**Table 1.** Butcher tableau of the explicit generic second order Runge-Kutta (RK) method, $p \in \mathbb{R}^+$.

| | | | | |
|---|---|---|---|---|
| 0 | | 0 | | 0 |
| $p$ | | $p$ | | $p$ |
| | | $1 - \frac{1}{2p}$ | | $\frac{1}{2p}$ |

The first attempt could be to replace the time step $h$ in these second order RK schemes by $\left[ 1 - \exp\left( -\frac{2\alpha h}{\Delta x^2} \right) \right] \frac{\Delta x^2}{2\alpha}$ to stabilize it. We obtained that although this replacement indeed

yields stable algorithms, they are still not second order. That is why we must return to the original logic by which we derived the first order CN method and combine it with the logic of the two-stage RK method above. At the first stage, we can calculate new "predictor" values of the variables, using (6) but with $h_1 = ph$ time step:

$$u_{\mathrm{i}}^{C_{\mathrm{P}}} = u_{\mathrm{i}}^{\mathrm{n}} \cdot e^{-\frac{ph}{\tau_{\mathrm{i}}}} + a_{\mathrm{i}} \tau_{\mathrm{i}} \cdot \left(1 - e^{-\frac{ph}{\tau_{\mathrm{i}}}}\right). \tag{9}$$

At the second stage, we can calculate the linear combination:

$$u_{\mathrm{i}}^{\mathrm{comb}} = \left(1 - \frac{1}{2p}\right) u_{\mathrm{i}}^{\mathrm{n}} + \frac{1}{2p} u_{\mathrm{i}}^{C_{\mathrm{P}}} \tag{10}$$

and take a full time step using these combinations as the more accurate constant values $u_{\mathrm{j}}^{\mathrm{comb}}$ of the neighbours towards which the $u_{\mathrm{i}}$ unknown functions are tending with increasing time step size $h$ because $e^{-\frac{ph}{\tau_{\mathrm{i}}}} \to 0$ if $h \to 0$. This means we have to replace $a_{\mathrm{i}}$ in (6) to:

$$a_{\mathrm{i}}^{\mathrm{comb}} = \sum_{\mathrm{j} \neq \mathrm{i}} m_{\mathrm{ij}} u_{\mathrm{j}}^{\mathrm{comb}} + Q_{\mathrm{i}}, \tag{11}$$

$$u_{\mathrm{i}}^{\mathrm{n}+1} = u_{\mathrm{i}}^{\mathrm{n}} \cdot e^{-\frac{h}{\tau_{\mathrm{i}}}} + a_{\mathrm{i}}^{\mathrm{comb}} \tau_{\mathrm{i}} \left(1 - e^{-\frac{h}{\tau_{\mathrm{i}}}}\right). \tag{12}$$

For a standard one dimensional homogeneous equidistant grid, we suggest the following Algorithm 1:

---

**Algorithm 1.** (Two-stage Constant-neighbour with ph time step and Constant-neighbour with full time step (briefly: CpC) method):

---

**Stage 1.** Take a $h_1 = ph$ time step with the CN method with $p \in \mathbb{R}^+$:
$$u_{\mathrm{i}}^{C_{\mathrm{P}}} = u_{\mathrm{i}}^{\mathrm{n}} \cdot e^{-\frac{2\alpha ph}{\Delta x^2}} + \left[\frac{u_{\mathrm{i}-1}^{\mathrm{n}} + u_{\mathrm{i}+1}^{\mathrm{n}}}{2} + \frac{\Delta x^2 Q_{\mathrm{i}}}{2\alpha}\right] \left(1 - e^{-\frac{2\alpha ph}{\Delta x^2}}\right).$$

**Stage 2.** Take a full $h$ time step with the CN method:
$$u_{\mathrm{i}}^{\mathrm{n}+1} = u_{\mathrm{i}}^{\mathrm{n}} \cdot e^{-\frac{2\alpha h}{\Delta x^2}} + \left[\frac{u_{\mathrm{i}-1}^{\mathrm{comb}} + u_{\mathrm{i}+1}^{\mathrm{comb}}}{2} + \frac{\Delta x^2 Q_{\mathrm{i}}}{2\alpha}\right] \left(1 - e^{-\frac{2\alpha h}{\Delta x^2}}\right)$$
where
$$u_{\mathrm{j}}^{\mathrm{comb}} = \left(1 - \frac{1}{2p}\right) u_{\mathrm{j}}^{\mathrm{n}} + \frac{1}{2p} u_{\mathrm{j}}^{C_{\mathrm{P}}}.$$

---

The $C\frac{1}{2}C$, $C\frac{2}{3}C$, $C1C$ versions are analogous to the (explicit) midpoint, Ralston and trapezoidal methods, respectively. In Section 3 we will analyse the general CpC algorithm and prove that it is second order in the time step size $h$. One might think that this whole procedure of constructing new methods using the already known RK schemes is trivial. To refute this opinion it is enough to note that following the logic explained above and using the Butcher-tableau of third order RK methods, the obtained methods are not third order, sometimes not even second order.

### 2.3. Extension to a General Grid

For a realistic system, the discretization must reflect the geometrical and material properties of the system; thus, we start from the more general Equation (2), where $\alpha, k, c$ and $\rho$ are not considered uniform in space. Using Formula (3) for a one dimensional, equidistant grid, we have

$$c(x)\rho(x)\frac{\partial u}{\partial t}\bigg|_{x} = \frac{1}{\Delta x}\left[k\left(x + \frac{\Delta x}{2}\right)\frac{u(x + \Delta x) - u(x)}{\Delta x} + k\left(x - \frac{\Delta x}{2}\right)\frac{u(x - \Delta x) - u(x)}{\Delta x}\right]$$
$$+ c(x)\rho(x)q(x).$$

Now we change to cell variables, where the indices refer to whole cells:

$$\frac{du_i}{dt} = \frac{A}{c_i \rho_i A \Delta x}\left(k_{i,i+1}\frac{u_{i+1} - u_i}{\Delta x} + k_{i-1,i}\frac{u_{i-1} - u_i}{\Delta x}\right) + Q_i \,.$$

Here $u_i$ is the temperature of the cell $i$, is the heat capacity of the cell in units $[J/K]$, $\Delta x$ is the length, $V = A\Delta$ is the volume while $m$ is the mass of the cell. We introduce two other quantities, the heat source term $Q$ of the cells:

$$Q_i = \frac{1}{V_i}\int_{V_i} q dV \,, \quad \text{in} \left[\frac{K}{s}\right] \text{ units,}$$

and the thermal resistance $R_{ij} = \frac{\Delta x}{k_{ij} A}$ in $[K/W]$ units. In the case of an irregular grid, the distances between the cell-centres are $d_{ij} = (\Delta x_i + \Delta x_j)/2$ and the resistances can be approximated as $R_{ij} \approx \frac{d_{ij}}{k_{ij} A_{ij}}$. Using these notations we have:

$$\frac{du_i}{dt} = \frac{u_{i-1} - u_i}{R_{i-1,i}\,C_i} + \frac{u_{i+1} - u_i}{R_{i+1,i}\,C_i} + Q_i \,.$$

This can be generalised easily to obtain the ODE system for a general (possibly unstructured) grid, which gives the time derivative of each temperature independently of any coordinate-system:

$$\frac{du_i}{dt} = \sum_{j \neq i}\frac{u_j - u_i}{R_{i,j}C_i} + Q_i \,.$$

Formally we can write this equation system into the same matrix-form as in (4). Now an off-diagonal $m_{ij} = 1/(R_{ij}C_i)$ element of the $M$ matrix can be nonzero only if the cells i and j are neighbours, i.e., there is direct heat conduction between them. Using this fact we can write the diagonal elements of the matrix as:

$$m_{ii} = \sum_{j \in \text{neighbours(i)}}\frac{-1}{R_{i,j}C_i} \doteq \frac{-1}{\tau_i} \,.$$

After this generalization of the system matrix and time constant $\tau$ we can immediately use not only the CN algorithm (6) but the Formulas (9)–(12) as well for the CpC method.

## 3. The Properties of the Methods for the Heat Equation

The new CN and CpC methods can be used to solve general ODE systems as well as the special ODE systems obtained by the semidiscretization of PDEs. In the following pages, we will consider both aspects, but we certainly do not state that they can be recommended for any kind of ODE systems and PDEs. First, we collect those properties of the methods which are obvious and do not require to be proved:

- They are explicit; we use matrices only to store data in a structured way, but do not perform operations with them and in the loop for the time steps, we use only vectors, not matrices. It also implies that the methods are easily parallelizable.
- They are one-step methods. When the $u_i^{n+1}$ values are calculated, only the values $u_j^n$ at the beginning of the current timestep are used. This implies that the methods are self-starting, the step size can be changed without any difficulty and the memory requirements are quite low.
- They can be easily applied for any space dimensions, unstructured grid or inhomogeneous heat conduction medium provided that the heat capacity of the cells and the thermal resistance between the cells can be calculated.

At this point, we have to underline again that these new methods may not be considered as Finite Difference Methods. Here time derivatives are not approximated by finite

difference formulas as in FDM. Because of the exponential terms, our methods might also seem to be similar to the so-called generalized Runge-Kutta Formula [40] or exponential integrators [41]. However, those methods use matrix exponentials, while we do not even need to use matrices during the calculation; thus, our methods are fundamentally different from those.

*3.1. Convergence*

In this section, we analyse the convergence properties of the methods as solvers for the spatially discretized systems. In Section 3.3 these will be examined regarding the methods as PDE solvers.

**Theorem 1.** *The new CpC method, defined by Equations (9)–(12), is a second order numerical method for arbitrary nonzero $p \in \mathbb{R}$ and for the general:*

$$\frac{d\vec{u}}{dt} = M\vec{u} + \vec{Q}, \quad \vec{u}(\text{t} = 0) = \vec{u}(0) \tag{13}$$

*linear ODE initial value problem, where M is an arbitrary nonsingular constant matrix, while $\vec{Q}$ and $\vec{u}(0)$ are arbitrary vectors.*

**Proof of Theorem 1.** To help the reader follow the argument, we present the matrix form of the ODE system:

$$\frac{d}{dt}\begin{pmatrix} u_1 \\ \vdots \\ u_N \end{pmatrix} = \begin{pmatrix} m_{11} & \dots & m_{1N} \\ \vdots & \vdots & \vdots \\ m_{N1} & \dots & m_{NN} \end{pmatrix}\begin{pmatrix} u_1 \\ \vdots \\ u_N \end{pmatrix} + \begin{pmatrix} Q_1 \\ \vdots \\ Q_N \end{pmatrix}.$$

The exact solution of the initial value problem (13) can be written as follows:

$$\vec{u}(t) = e^{Mt}\vec{u}(0) + \left(e^{Mt} - 1\right)M^{-1}\vec{Q}$$
$$= \left(1 + Mt + M^2\frac{t^2}{2} + M^3\frac{t^3}{3!} + \dots\right)\vec{u}(0) + \left(t + M\frac{t^2}{2} + M^2\frac{t^3}{3!} + \dots\right)\vec{Q}.$$

Without the loss of generality we examine only $u_1$ at the first time step. It implies that we use the initial values $u_j(0)$ at the beginning of the time step. The $0^{th}$ and first order terms in the exact solution at $t = h$ are the following

$$u_1(h) = u_1(0)(1 + m_{11}h) + h\sum_{j>1}m_{1j}u_j(0) + Q_1h. \tag{14}$$

The second order terms:

$$u_1(h) = \frac{h^2}{2}\sum_{j=1}^N m_{1j}\sum_{k=1}^N m_{jk}u_k(0) + \frac{h^2}{2}\sum_{j=1}^N m_{1j}Q_j. \tag{15}$$

When we apply the CpC method for the general initial value problem (13), at the first stage we obtain:

$$u_1^{\text{Cp}}(ph) = u_1(0)e^{m_{11}ph} + \frac{a_1}{m_{11}}\left(e^{m_{11}ph} - 1\right)$$

where

$$a_1 = \sum_{j>1}m_{1j}u_j(0) + Q_1.$$

The result of the second stage in the CpC method is the following:

$$u_1^{\text{CpC}}(h) = u_1(0)e^{m_{11}h} - \frac{a_1^{\text{comb}}}{m_{11}}\left(1 - e^{m_{11}h}\right).$$

The first term, up to second order is simple:

$$u_1(0)\left(1 + m_{11}h + \frac{m_{11}^2 h^2}{2} + \dots\right). \tag{16}$$

To examine the second term we have to do the substitutions using (11) and (10):

$$-\frac{1}{m_{11}}\left(\sum_{j>1} m_{1j}u_j^{\text{comb}} + Q_1\right)\left(1 - e^{m_{11}h}\right)$$

$$= -\frac{1}{m_{11}}\left(\sum_{j>1} m_{1j}\left[\left(1 - \frac{1}{2p}\right)u_j(0) + \frac{1}{2p}u_j^{\text{pred}}\right] + Q_1\right)\left(1 - e^{m_{11}h}\right)$$

$$= -\frac{1}{m_{11}}\left(\sum_{j>1} m_{1j}\left[\left(1 - \frac{1}{2p}\right)u_j(0) + \frac{1}{2p}\left\{u_j(0)e^{m_{jj}ph} + \frac{1}{m_{jj}}\left(\sum_{k\neq j} m_{jk}u_k(0) + Q_j\right)\left(e^{m_{jj}ph} - 1\right)\right\}\right] + Q_1\right) \times \left(1 - e^{m_{11}h}\right)$$

which can be written as:

$$\left(\sum_{j>1} m_{1j}\left[\left(1 - \frac{1}{2p}\right)u_j(0) + \frac{1}{2p}\left\{\begin{array}{l} u_j(0)\left(1 + m_{jj}ph + \frac{(m_{jj}ph)^2}{2} + \dots\right) + \frac{1}{m_{jj}}\times \\[2mm] \times\left(\sum_{k\neq j} m_{jk}u_k(0) + Q_j\right)\left(m_{jj}ph + \frac{(m_{jj}ph)^2}{2} + \dots\right) \end{array}\right\}\right] + Q_1\right) \times \frac{\left(m_{11}h + \frac{(m_{11}h)^2}{2} + \dots\right)}{-m_{11}}.$$

As the last term contains terms at least first order, we can omit the second order terms in the large bracket and obtain:

$$\left(\sum_{j>1} m_{1j}u_j(0)\left(1 - \frac{1}{2p} + \frac{1 + m_{jj}ph}{2p} + \dots\right) + \frac{1}{2p}\sum_{j>1} m_{1j}\left(\sum_{k\neq j} m_{jk}u_k(0) + Q_j\right)ph + Q_1\right)\left(h + m_{11}\frac{h^2}{2} + \dots\right)$$

$$+ O(h^3) = h\sum_{j>1} m_{1j}u_j(0) + Q_1 h + \frac{h^2}{2}\sum_{j>1} m_{1j}m_{jj}u_j(0) + \frac{h^2}{2}m_{11}\sum_{j>1} m_{1j}u_j(0) \tag{17}$$

$$+ \frac{h^2}{2}\sum_{j>1} m_{1j}\sum_{k\neq j} m_{jk}u_k(0) + \frac{h^2}{2}\sum_{j>1}^{N} m_{1j}Q_j + \frac{h^2}{2}m_{11}Q_1 + O(h^3) \quad.$$

Notice that the parameter $p$ has been completely cancelled out. Now we can easily see that the zeroth and first order terms in (16) and (17) together are the same as in (14). Let us collect the second order terms:

$$\frac{h^2}{2}\left[\underbrace{u_1(0)m_{11}^2 + m_{11}\sum_{j>1} m_{1j}u_j(0)}_{m_{11}\sum_{j=1}^{N} m_{1j}u_j(0)} + \underbrace{\sum_{j>1} m_{1j}m_{jj}u_j(0) + \sum_{j>1} m_{1j}\sum_{k\neq j} m_{jk}u_k(0)}_{\sum_{j>1} m_{1j}\sum_{k=1}^{N} m_{jk}u_k(0)} + \underbrace{m_{11}Q_1 + \sum_{j>1} m_{1j}Q_j}_{\sum_{j=1}^{N} m_{1j}Q_j}\right]$$

$$\underbrace{\qquad\qquad\qquad\qquad\qquad\qquad\qquad\qquad\qquad\qquad\qquad\qquad}_{\sum_{j=1}^{N} m_{1j}\sum_{k=1}^{N} m_{jk}u_k(0)}$$

which is equivalent to (15), i.e., the CpC method is (at least) 2nd order.  □

### 3.2. Stability

We recall that when the stability of the ODE solvers are examined, typically the following scalar equation is used

$$\frac{du}{dt} = -\lambda u, \ \lambda \geq 0.$$

Absolute stability means that arbitrarily large time steps yield a bounded solution for this equation [42] and [43] (p. 13). It is well-known that the stability function of any explicit

Runge–Kutta method is a polynomial, so they can never be A-stable, which limits their use for solving PDEs ([43], p. 24). Now, for this scalar equation, i.e., in the case of one cell, our methods can be defined only in one meaningful way: To provide the analytical solution. This means that in this sense, they are trivially A-stable. We therefore examine the stability when $u$ is a vector, as it is the case for spatially discretized PDEs. Unlike Runge-Kutta methods, Formulas (6), (9) and (12) use an exponential expression of the time step size $h$ to calculate the new values of the variable $u$. At the first stage, Equation (9) or Stage 1 in Algorithm 1 the temperature $u_i$ of each cell tends to $a_i \tau_i$ with increasing $h$, which is the weighted average of the temperatures of the neighbouring cells at the beginning of the time step. At the second stage, $u_i$ tends to the combination of the beginning temperatures of the neighbours and the neighbours of the neighbours with increasing $h$. We will formulate this statement in Theorem 2, but first we evoke the following simple lemma, the associativity of convex combinations ([44], p. 28) on which we build during the proof of the theorem.

**Lemma 1.** *A convex combination Lemma: A convex combination $x = \sum a_i x_i$ of convex combinations $x_i = \sum b_{ij} y_{ij}$ is still a convex combination:*

$$x = \sum \sum \left( a_i b_{ij} \right) y_{ij}$$

*for any $y_{ij} \in \mathbb{R}^n$.*

**Theorem 2.** *In the case of the heat equation, if the $Q_i$ source terms are zero, the new values $u_i^{CpC}$ are the convex combination of the initial values $u_j(0)$, $j = 1, ..., N$ for any $p \geq \frac{1}{2}$.*

**Proof of Theorem 2.** We will use the facts that $\tau_i = -1/m_{ii}$ and $m_{ij,j\neq i}$ are non-negative quantities; thus, $0 < e^{-h/\tau_i} = e^{m_{ii}h} \leq 1$ holds because of physical reasons, namely the second law of thermodynamics. Again, without the loss of generality, we examine only $u_1$ at the first time step.

(A) Let us examine the first stage:

$$u_1^{Cp} = u_1(0) \cdot e^{-\frac{ph}{\tau_1}} + a_1 \tau_1 \cdot \left( 1 - e^{-\frac{ph}{\tau_1}} \right). \tag{18}$$

The term $a_1 \tau_1$ is the following:

$$a_1 \tau_1 = \frac{-1}{m_{11}} \sum_{j>1} m_{ij} u_j(0) .$$

Let us calculate the coefficients of the $u_j(0)$ initial values:

$$\frac{-m_{1j}}{m_{11}} = \frac{1}{C_1 R_{1j}} \frac{1}{\sum_{k>1} \frac{1}{C_1 R_{1k}}} = \frac{\frac{1}{R_{1j}}}{\sum_{k>1} \frac{1}{R_{1k}}} .$$

One can see that they are nonnegative and their sum is one, which implies that $a_1 \tau_1$ is a convex combination $u_j(0)$. Now it is clear that the coefficients $e^{-\frac{ph}{\tau_1}}$ and $1 - e^{-\frac{ph}{\tau_1}}$ in (18) are also nonnegative and their sum is one, and therefore $u_1^{Cp}$ is also a convex combination of the initial values $u_j(0)$, $j = 1, ..., N$.

(B) The final solution at the end of the first time step is:

$$u_1^{CpC} = u_1(0) \cdot e^{-\frac{h}{\tau_1}} + a_1^{comb} \tau_1 \cdot \left( 1 - e^{-\frac{h}{\tau_1}} \right),$$

where

$$a_1^{\text{comb}} \tau_1 = \frac{-1}{m_{11}} \sum_{j>1} m_{1j} \left[ \left( 1 - \frac{1}{2p} \right) u_j(0) + \frac{1}{2p} u_j^{\text{Cp}} \right] .$$

The square bracket is a convex combination if $0 \leq \frac{1}{2p} \leq 1$ and $0 \leq 1 - \frac{1}{2p} \leq 1$ , which holds if $\frac{1}{2} \leq p$. As we have seen at point A) the factors $\frac{-m_{1j}}{m_{11}}$ are nonnegative and their sum is one; thus, $u_1^{\text{C}p\text{C}}$ is a convex combination of the initial values. $\square$

Theorem 2 implies that any error produced during the calculation will be distributed quickly and cannot be amplified; thus, the methods are stable for the heat equation (and as we will show later, for Fisher's equation). Moreover, in the homogeneous ($Q = 0$) case, Theorem 2 guarantees that the solution follows the maximum and minimum principles ([43], p. 87), i.e., the extreme values of $u$ occur among the initial or the prescribed boundary values. This means that starting from positive initial values and assuming no heat sinks are present, the variables always remain positive. This property is especially advantageous when Equations (1) and (2) are used to model physical processes, where the temperature measured in Kelvin (or analogous variables such as concentrations of chemical species) cannot be negative.

### 3.3. Analysis of the Errors

We analytically calculated the first two terms of the error of the CpC scheme for a system of 2 identical cells. At the end of the first time step, the error for the first cell is the following:

$$u_1^{\text{exact}}(h) - u_1^{\text{num}}(h) = \frac{1+p}{4\tau^3}(u_2(0) - u_1(0))h^3 + \frac{1+3p}{12\tau^2}Q_2 h^3 + \frac{1}{6\tau^2}(Q_1 - Q_2)h^3 + O\left(h^4\right) . \tag{19}$$

We note that we obtained similar expressions for larger systems and for the higher order error terms as well, but of course with more terms and with larger exponents. We can draw two conclusions.

First, for not too large time steps, the error is smaller for smaller parameter values $p$, so $p$ should be chosen as small as possible. Unfortunately, this advice contradicts the requirement of stability, which is not guaranteed for $p < 0.5$. We will return to this question during the discussion of the numerical experiments.

For the second consequence we recall that $\tau \sim \Delta x^2$ , which means that for small $\Delta x$ , the main error terms contain powers of $\frac{h}{\Delta x^2}$ . We have to conclude that the methods are only conditionally consistent in the sense that when the size of the spatial cells $\Delta x$ tends to zero, the time step size $h$ also has to go to zero faster than the second power of $\Delta x$ . If $\Delta x \to 0$ while $h/(\Delta x)^2$ is kept constant, then the global errors tend to some constant values, and if one decreases $h$ faster, the errors tend to zero. However, if $h$ is decreased slower, the errors are growing, but not without limits: They approach their maximum possible values which are restricted by the above mentioned maximum and minimum principles. It implies that considering as solvers for the original PDE, the methods are conditionally convergent. However, as we have proved above, they are unconditionally convergent for any fixed spatial grid, i.e., the solution converges to the exact solution of the ODE system obtained by the discretization of space variables when the time step size $h$ tends to zero. It is worth mentioning that this second consequence holds for the CN method as well.

## 4. Comparison with Exact Results

We note that all simulations are performed using the MATLAB R2019b software on a desktop computer with an Intel Core i3-8100 as CPU.

*4.1. The Heat Equation*

We solve PDE (1) on the interval $x \in [0,1]$ with the following heat source term:

$$q(x,t) = q_0 \sin(\pi x),$$

where $q_0 = 2$ and $\alpha = 1$. The initial condition is also a sine function but with smaller wavelength:

$$u(x,t=0) = \sin(3\pi x)$$

and we use zero Dirichlet boundary conditions:

$$u(x=0,t) = u(x=1,t) = 0.$$

It is easy to check that the analytical solution of this problem is:

$$u(x,t) = \frac{q_0}{\pi^2} \sin(\pi x)\left(1 - e^{-\pi^2 t}\right) + \sin(3\pi x)e^{-9\pi^2 t}.$$

We discretize $x$ by setting:

$$x_j = j\Delta x, \ j = 0, ..., 100, \quad \Delta x = 0.01.$$

The solutions are examined and compared at final time $t_{fin} = 0.2$. We define the (global) error as the maximum of the absolute value of the difference between the exact temperature $u_i^{exact}$ and the temperature $u_i^{num}$ obtained by our numerical methods at the end of the examined time interval:

$$Error = \max_i \left| u_i^{exact}(t_{fin}) - u_i^{num}(t_{fin}) \right|. \tag{20}$$

In Figure 1, one can see these global errors as a function of time step size $h$. One can see that the error tends to a small nonzero value in all cases. This non-vanishing error is caused by the discretization of the space variable. More precisely, it is the truncation errors ([43], p. 7)

$$\varepsilon_i = -\frac{\Delta x^2}{12} f''''(\eta_i) \tag{21}$$

of the central difference Formula (3) accumulated during the subsequent time steps. In Figure 2, we illustrate the behaviour of the maximum errors for different space steps and for $p = 0.5$.

The minimum of these functions appears in the form of a cusp in each case. The reason for this is that the largest error terms (19) and (21), coming from the time-integration method and the spatial discretization, respectively, are of opposite signs and cancel each other at these special, optimal time step size and grid spacing combinations. Unfortunately, we observed this behaviour only when the source term $Q$ strongly dominates the effect of the (non-uniform) initial conditions. Moreover, the location of the cusp, i.e., the optimal $h$ depends on so many factors that this favourable property can hardly be exploited in practical problems.

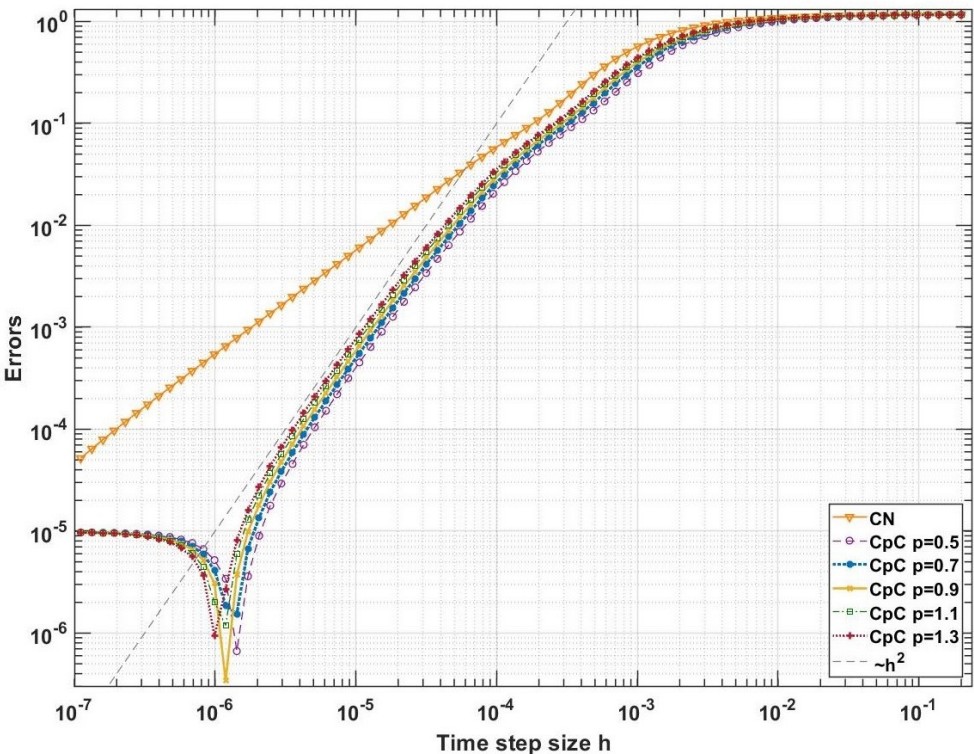

**Figure 1.** Maximum errors defined by formula (20) as a function of the time step size for our constant-neighbour (CN) and two-stage CpC methods in the case of the heat equation.

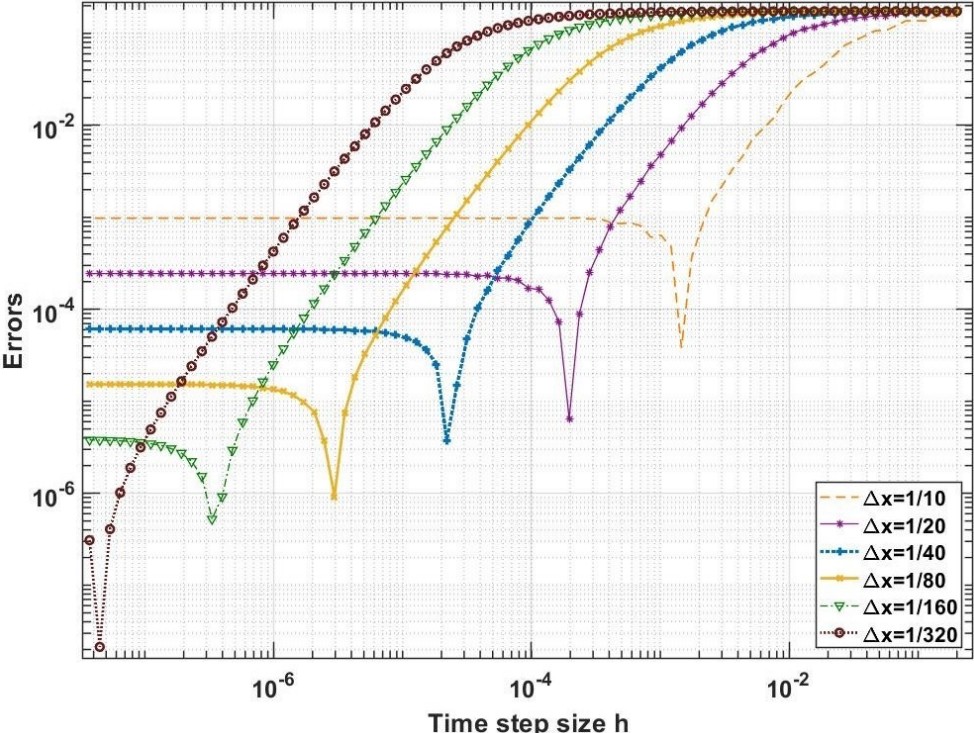

**Figure 2.** Maximum errors defined by Formula (20) as a function of the time step size in the case of our different grid spacings $\Delta x$ for the two-stage combined C½C methods applied for the heat equation.

*4.2. The Nonlinear Fisher's Equation*

We consider the following equation

$$\frac{\partial u}{\partial t} = \alpha \nabla^2 u + \beta u(1 - u) \tag{22}$$

with $\alpha = 1$ and $\beta > 0$, subject to the following initial condition:

$$u(x, t = 0) = \left(1 + e^{\sqrt{\frac{\beta}{6}}x}\right)^{-2}.$$

The analytical solution of this problem is the following travelling wave function [45,46]

$$u^{\text{exact}}(x, t) = \left(1 + e^{\sqrt{\frac{\beta}{6}}x - \frac{5}{6}\beta t}\right)^{-2}.$$

The appropriate Dirichlet boundary conditions are prescribed at the two ends of the interval:

$$u(x = x_0, t) = \left(1 + e^{\sqrt{\frac{\beta}{6}}x_0 - \frac{5}{6}\beta t}\right)^{-2}, \quad u(x = x_{\text{fin}}, t) = \left(1 + e^{\sqrt{\frac{\beta}{6}}x_{\text{fin}} - \frac{5}{6}\beta t}\right)^{-2}.$$

The values of the exact solution obviously lie in the unit interval, $u(x, t) \in [0, 1]$, $x, t \in \mathbb{R}$. We take the nonlinear term into account in a "semi-implicit" way:

$$u_i^{n+1} = u_i^{n+1,\text{pred}} + \beta u_i^{n+1,\text{pred}}\left(1 - u_i^{n+1}\right)h.$$

where $u_i^{n+1,\text{pred}}$ is the result of the first and the second stage using the same formulas as in the previous section, i.e., with taking into account the effect of the diffusion term only. Note that now the new value $u_i^{n+1}$ appeared on the right hand side of the equation as well. However, this equation can be rearranged into a fully explicit form:

$$u_i^{n+1} = \frac{1 + \beta h}{1 + \beta h u_i^{n+1,\text{pred}}} u_i^{n+1,\text{pred}}. \tag{23}$$

This operation is performed in a separate third stage after Stage 2 (a separate second stage after the first one in case of the CN method), where a loop is going through all the nodes. We have done several tests with this semi-implicit treatment and obtained that the methods behave similarly well as in the linear case. The errors as a function of the time step size $h$ are presented in Figure 3 for $\beta = 2.5$, $x_0 = 0$, $x_{\text{fin}} = 4$, $t_{\text{fin}} = 2$, and $\Delta x = 0.01$.

**Remark 1.** *Suppose that $0 \leq u_i^{n+1,\text{pred}} \leq 1$. Now from Equation (23), it is obvious that the new value $u_i^{n+1}$ cannot be negative, and also cannot exceed 1; thus, $0 \leq u_i^{n+1} \leq 1$.*

**Corollary 1.** *In the case of Fisher's equation (22) if the initial values are in the interval $u_i^0 \in [0, 1]$, then the values of u remain in this interval for arbitrary $\beta$ and time step size h, for the whole calculation. This is a special case of the maximum and minimum principles, which implies that the CN and the CpC methods supplemented with (23) is unconditionally stable for Fisher's equation.*

**Proof.** Theorem 2 and Remark 1 immediately imply the statement.　□

However, we do not state that we have found the optimal algorithm to solve Fisher's equation. The purpose of this subsection is to demonstrate that these methods can be successfully applied for nonlinear equations as well. The systematic investigation of the application possibilities of the methods for nonlinear equations is clearly out of the scope of this paper and planned to be published in a series of papers in the future.

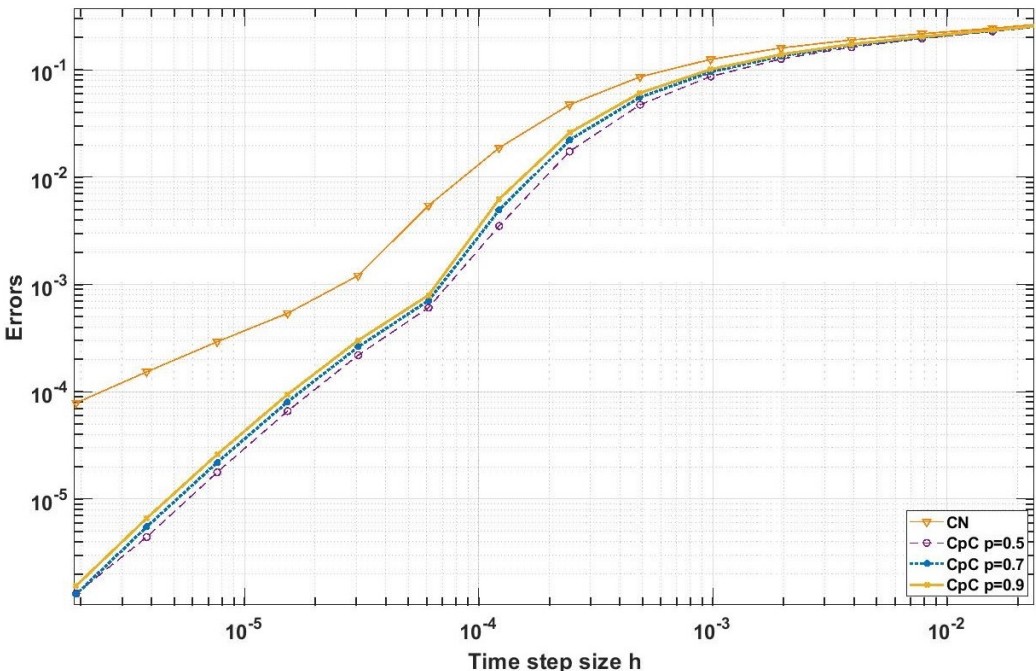

**Figure 3.** Maximum errors defined by Formula (23) as a function of the time step size in the case of our constant-neighbour (CN), the two-stage CpC methods for the nonlinear Fisher's equation.

## 5. Comparison with Numerical Results

In this section rectangle-shaped lattices are examined with $N = N_x \times N_z$ cells, where the edge of the system is thermally isolated regarding conductive type heat transfer, which means closed (zero Neumann) boundary conditions. To help the reader to imagine the system, we present the arrangement of the variables in Figure 4. We emphasize that the size and the shape of the cells are not necessarily identical.

**Figure 4.** Arrangement of the examined system. The outer double line represents thermal isolation and the inner thin lines represent cell borders. The red double arrows symbolize conductive heat transport through the resistances $R_{ij}$ while $Q_N$ is an external heat source.

To obtain a reference solution, we used the implicit `ode15s` solver of MATLAB, which is a variable-step, variable-order solver based on the numerical differentiation formulas (NDFs) of orders 1 to 5, where the letter s indicates that the codes were suggested to use in the case of stiff systems. During this calculation, we applied stringent error tolerance ('RelTol' and 'AbsTol' were both $10^{-16}$). We compare the performance of the new CN and CpC methods with professionally coded and widely used MATLAB routines. In addition to `ode15s`, the following solvers are employed, with loose error tolerance to increase the speed:

- `ode45`, the explicit Runge–Kutta–Dormand–Prince formula of order 4(5).
- `ode23`, the explicit Runge–Kutta–Bogacki–Shampine method of order 2(3).

- `ode23t`, an implementation of the trapezoidal rule using a "free" interpolant .
- `ode23s`, a modified Rosenbrock formula of order 2.

Besides the above listed MATLAB solvers, we have coded the previously mentioned unconditionally positive finite-difference (UPFD) method of Chen–Charpentier et al. [27] for comparison purposes because of its simplicity. Applying that scheme to (1) we obtain:

$$\frac{u_i^{n+1} - u_i^n}{h} = \frac{\alpha}{\Delta x^2}\left(u_{i-1}^n - 2u_i^{n+1} + u_{i+1}^n\right) + Q_i \, .$$

This formula can be rewritten in the following explicit form:

$$u_i^{n+1} = \left(\frac{u_i^n}{h} + \frac{\alpha}{\Delta x^2}\left(u_{i-1}^n + u_{i+1}^n\right) + Q_i\right)\left(\frac{1}{h} + \frac{2\alpha}{\Delta x^2}\right)^{-1}$$

which can be generalized in the manner we explained in Section 2.3, so the UPFD formula we use:

$$u_i^{n+1} = \frac{\frac{u_i^n}{h} + \sum_{j \neq i} \frac{u_j^n}{C_i R_{ij}} + Q_i}{\frac{1}{h} + \frac{1}{\tau_i}} \, . \tag{24}$$

Here we define the (global) error in the same way as in (20) but now we have a reference solution instead of the exact analytical solution:

$$Error = \max_i \left| u_i^{\text{ref}}(t_{\text{fin}}) - u_i^{\text{num}}(t_{\text{fin}}) \right| \, . \tag{25}$$

### 5.1. First Case: A Very Stiff System

We set $N_x = 100$, $N_z = 40$; thus, the number of cells is $N = 4000$. Different random values are given to the heat capacities and to the thermal resistances:

$$C_i = 10^{(2-4 \times rand)}, \quad R_{x,i} = 10^{(3-6 \times rand)}, \quad R_{z,i} = 10^{(3-6 \times rand)}$$

where rand is a random number generated by the MATLAB uniformly in the (0,1) interval for each quantity. This means that the capacities (the resistances) follow a log-uniform distribution between 0.01 and 100 (between 0.001 and 1000). The initial temperatures have a uniform distribution between 0 and 1, i.e., $u_i(0) = rand$, while the source-terms have a uniform distribution between $-0.5$ and $0.5$. Starting from $t_0 = 0s$ we solve this system for the temperatures at final time $t_{fin} = 1s$.

As the system is thermally isolated, the system matrix has a zero eigenvalue, all other eigenvalues are negative. If we denote the eigenvalues with the largest (smallest) absolute value with $\lambda_{\text{MAX}}$ ($\lambda_{\text{MIN}}$), then the stiffness ratio can be calculated as

$$\frac{\lambda_{\text{MAX}}}{\lambda_{\text{MIN}}} = 1.8 \times 10^9 \, .$$

For the explicit Euler method, the maximum possible time step size is:

$$h_{\text{MAX}}^{\text{EE}} = \left| \frac{2}{\lambda_{\text{MAX}}} \right| = 1.3 \times 10^{-5} \text{ s} \, .$$

above this threshold instability necessarily appears. In Figure 5, we present the errors for the one-stage CN and the two-stage CpC methods in the case of different $p$ parameters as a function of the step size $h$. From the figure one can see that for small $h$, the global error decreases with the first power of the step size in the case of the Constant-neighbour methods and with the second power for the CpC methods, which confirms that the methods are first and second order, respectively. It can be seen that for $p = 1/3$ the method is unstable in the region of large step sizes, but in the region of small $h$-s it is slightly more accurate than for

larger $p$-s. For $p = 1/2$ the error-function has a relative elevation around $h = 10^{-2}...10^{-3}$, because this parameter value is at the border of the stability region.

We also examine the global errors as a function of the total running times. The results are presented in Figure 6. For smaller step sizes the running times of several runs have been averaged to avoid random fluctuations due to very short running times.

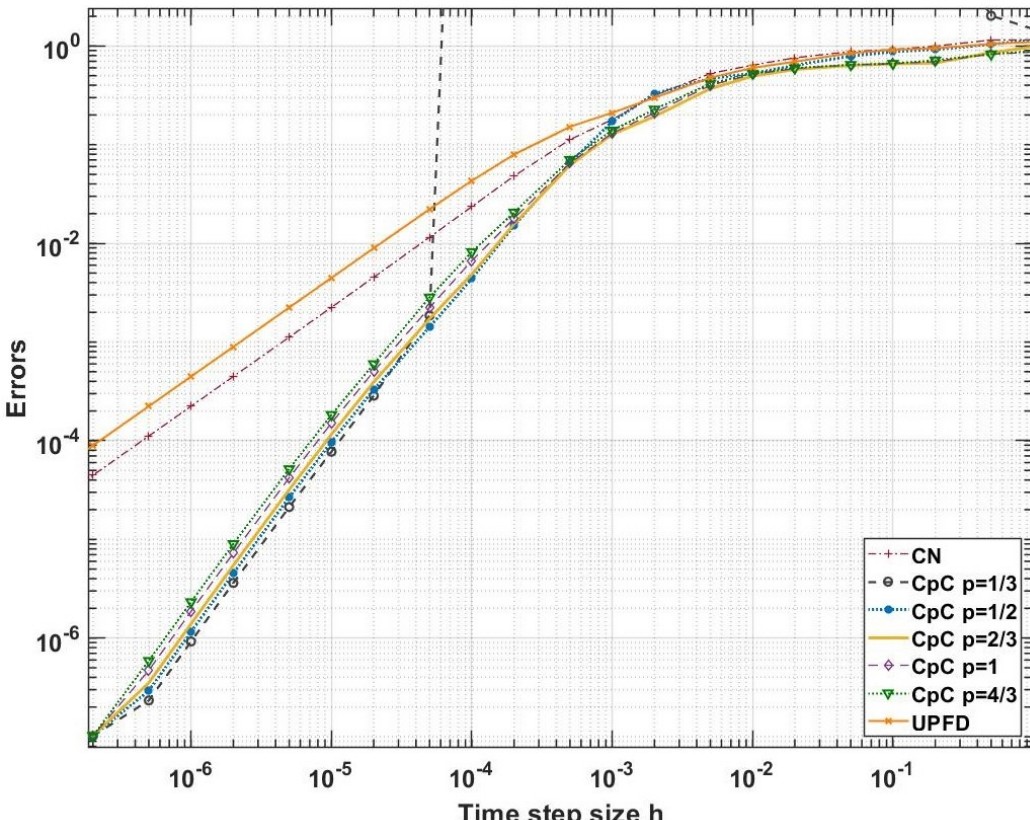

**Figure 5.** Maximum error defined by Formula (25) as a function of the time step size $h$ in the case of our constant-neighbour (CN), the two-stage CpC methods and the UPFD method defined by (22) applied for the system consisting of $N = 4000$ cells described in Section 5.1.

We note that the explicit solvers ode23 and ode45 can hardly provide any meaningful results, because for large error tolerance they diverge, while tightening the tolerance yields an error message "array exceeds maximum array size".

In Table 2 we summarize some results obtained by MATLAB routines ode15s, ode23s, ode23, ode23t, the UPFD scheme, and our methods $C_{1/2}C$ and $C_{2/3}C$. One can see that, if we set the same moderate precision requirements, our methods are much faster than the conventional explicit or implicit methods, even without adaptive stepsize control, parallelization, or vectorization. The UPFD method is also very fast, but its accuracy is far from being optimal.

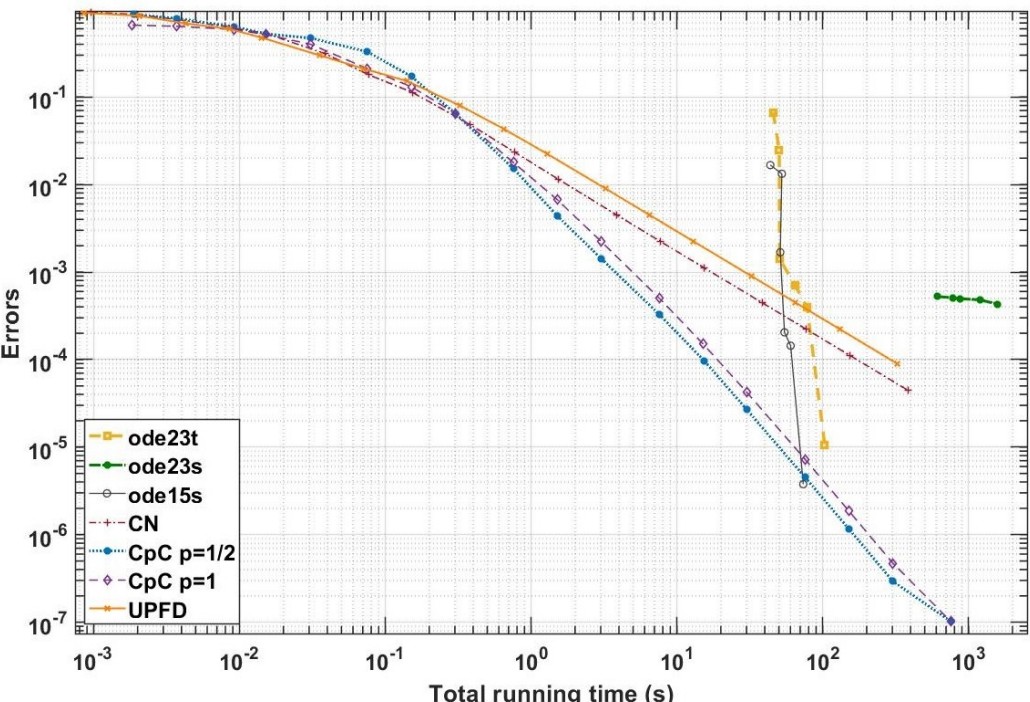

**Figure 6.** Maximum errors defined by Formula (25) as a function of the total running times in the case of our constant-neighbour (CN), the two-stage CpC methods with different values of parameter $p$, three different MATLAB solvers and the UPFD method applied for the system consisting of $N = 4000$ cells described in Section 5.1.

**Table 2.** Performance of different solvers for the system consisting of $N = 4000$ cells described in Section 5.1. CpC means the new two-stage method. The absolute and the relative tolerance 'RelTol' and 'AbsTol' have been set to the same values denoted by 'tol'. The error is defined by (25).

| Method | Running Time (s) | Error |
|---|---|---|
| ode15s, $tol = 200$ | 43.6 | 0.017 |
| ode15s, $tol = 0.03$ | 51.1 | 0.0017 |
| ode23s, $tol = 200$ | 608 | $5.2 \times 10^{-4}$ |
| ode23, $tol = 0.5$ | 2039 | 0.496 |
| ode23t, $tol = 200$ | 45.7 | 0.067 |
| ode23t, $tol = 1$ | 50.5 | 0.0014 |
| $C_{2/3}C$, $h = 5 \times 10^{-4}$ | 0.3 | 0.06 |
| $C_{1/2}C$, $h = 2 \times 10^{-5}$ | 7.55 | $3.3 \times 10^{-4}$ |
| $C_{1/2}C$, $h = 5 \times 10^{-6}$ | 30.2 | $2.7 \times 10^{-5}$ |
| UPFD, $h = 10^{-5}$ | 6.51 | 0.0045 |

### 5.2. Second Case: A Highly Anisotropic System

Now $N_x = 100$, $N_z = 100$; thus, $N = 10,000$. The capacities and the thermal resistances followed a log-uniform distribution again:

$$C_i = 10^{(1-2rand)}, \quad R_{x,i} = 10^{(3-2rand)}, \quad R_{y,i} = 10^{(-1-2rand)}.$$

Note that, on average, there is four orders of magnitude difference between the resistances in the $x$ and in the $z$ direction. The initial temperatures are $u_i(0) = 2rand - 1$ while the source-terms are $Q_i = rand - 0.5$. The task is to solve this initial value problem for the temperatures at $t_{fin} = 1s$ starting from $t_0 = 0$ s.

The stiffness ratio is rather high, $2.15 \times 10^9$. The maximum possible time step size for the explicit Euler method is $h_{MAX}^{EE} = 9.6 \times 10^{-5}$ s, a few times larger than in the first case.

In Figure 7, we present the error for the CN and the CpC methods as a function of the step size $h$.

We also present the global errors as a function of the total running times in Figure 8. In this case, due to the larger $h_{MAX}^{EE}$, the explicit solvers ode23 and ode45 are able to produce some results, albeit still in a relatively narrow region of tolerance values 'RelTol' and 'AbsTol'.

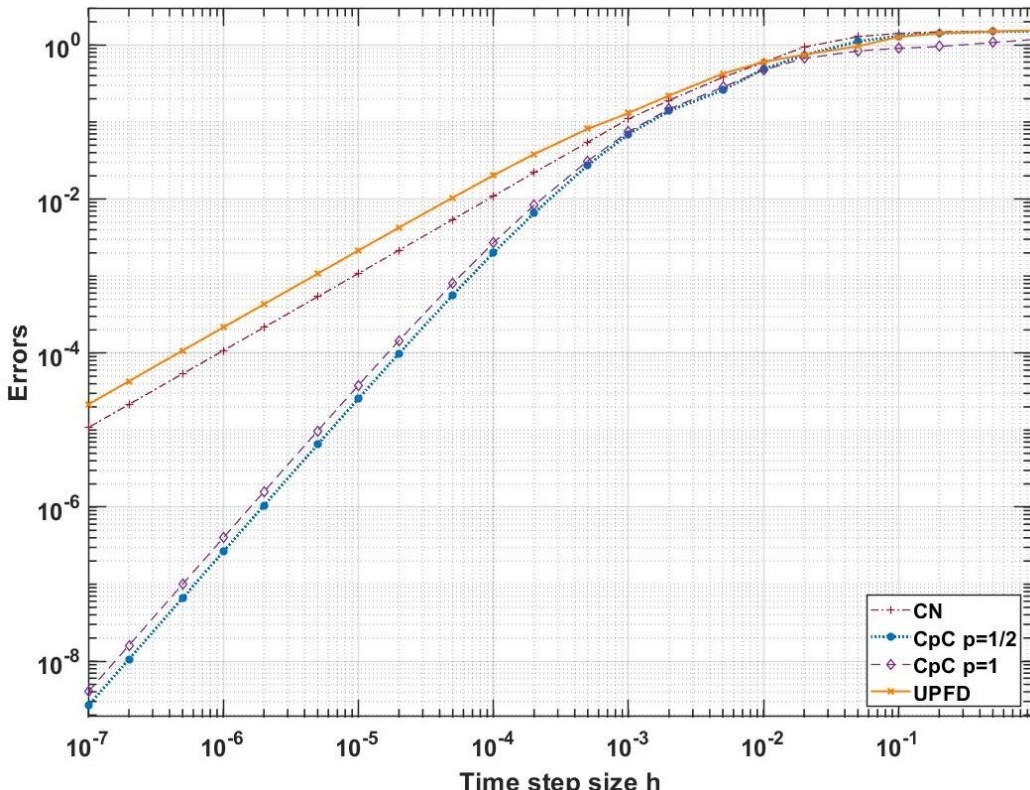

**Figure 7.** Maximum errors defined by Formula (25) as a function of the time step size in the case of our constant-neighbour (CN), the two-stage CpC methods, and the UPFD method applied for the system consisting of $N = 10,000$ cells described in Section 5.2.

One can see that our methods are much faster than the conventional explicit or implicit methods. One might think that the two-stage C½C method produces too large errors for medium step sizes, e.g., for $h = 5 \times 10^{-4}$. However, in this case the average absolute error is only 0.002 and from Figure 9 one can see that the graph of the produced solution-function is almost indistinguishable from the reference curve. This means that our methods can produce acceptable results three orders of magnitude faster than the well-established MATLAB routines.

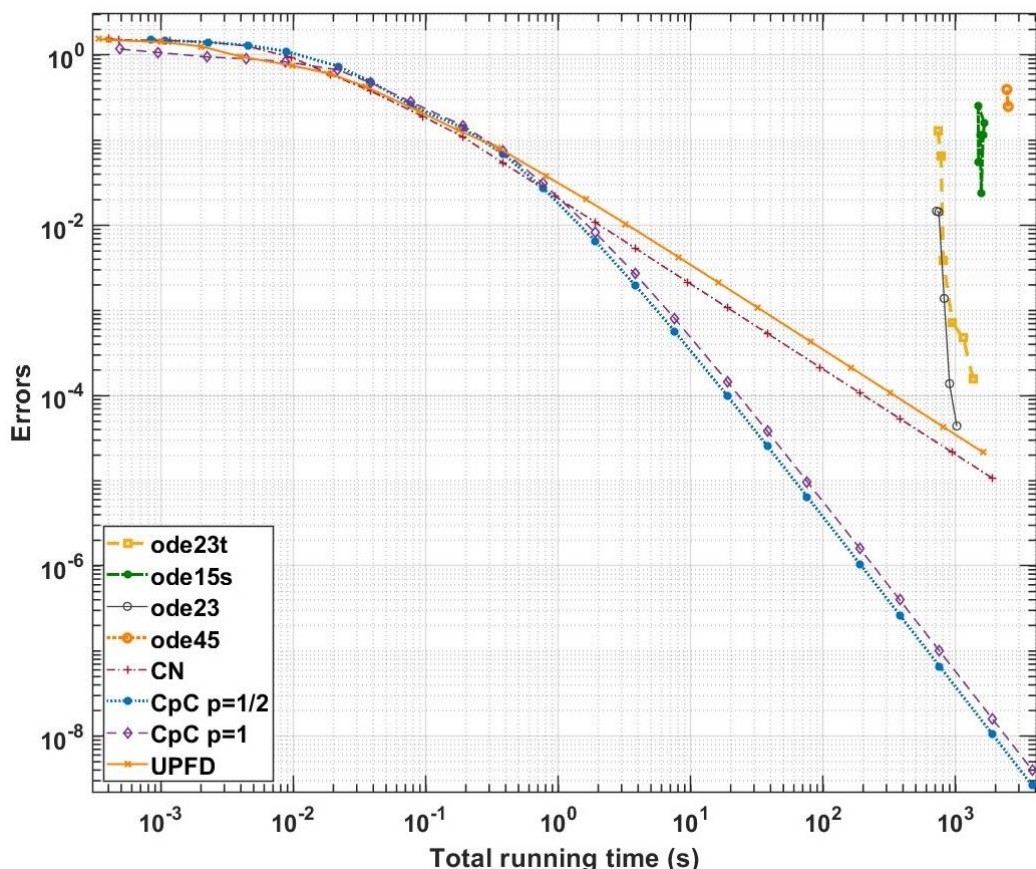

**Figure 8.** Maximum errors defined by Formula (25) as a function of the total running time in the case of our constant-neighbour (CN), the two-stage CpC methods, three different MATLAB solvers and the UPFD method applied for the system consisting of $N = 10,000$ cells described in Section 5.2.

In Table 3, we summarize some results obtained by MATLAB solvers ode23, ode45, ode15s, ode23s, ode23t, the UPFD scheme, and our two-stage CpC methods.

**Table 3.** Performance and errors of different solvers for the system consisting of $N = 10,000$ cells. For other notations, see caption of Table 2.

| Method | Running Time (s) | Error |
|---|---|---|
| ode15s, $tol = 200$ | 716 | 0.014 |
| ode15s, $tol = 0.03$ | 898 | $1.4 \times 10^{-4}$ |
| ode23s, $tol = 1$ | 11,893 | $5 \times 10^{-4}$ |
| ode23, $tol = 0.9$ | 1476 | 0.056 |
| ode23t, $tol = 2$ | 804 | 0.004 |
| ode45, $tol = 2.5$ | 2491 | 0.25 |
| $C_{1/2}C, h = 5 \times 10^{-4}$ | 0.76 | 0.028 |
| $C_{1/2}C, h = 2 \times 10^{-5}$ | 18.9 | $9.9 \times 10^{-5}$ |
| $C_{1/2}C, h = 10^{-6}$ | 379 | $2.6 \times 10^{-7}$ |
| UPFD, $h = 2 \times 10^{-5}$ | 8.04 | 0.0042 |

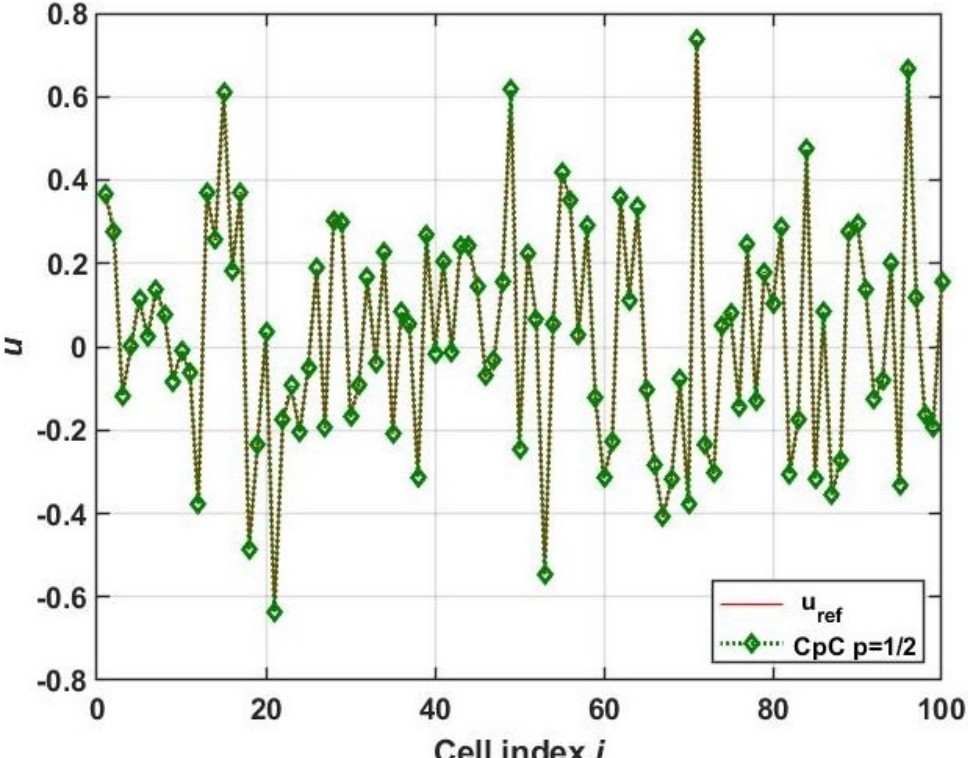

**Figure 9.** The variable $u$ as a function of the cell index i for the first row (first 100 cells, see Figure 4) in the case of the system consisting of $N = 10{,}000$ cells described in Section 5.2. The red continuous line represents the high-precision solution while the green dotted line with diamond markers are the values produced by our C½C algorithm for $h = 5 \times 10^{-4}$ in 0.76 s, orders of magnitude faster than the conventional solvers.

We emphasize that the $N = 10{,}000$ number of cells is still much smaller than in many real-world applications. Since for larger systems, implicit methods have a more serious disadvantage, we think that our methods are even more competitive in those cases. We note that the reason we have not used a larger number of cells for testing purposes is that our computer cannot solve them using the MATLAB routines because of the too large memory requirements.

## 6. Conclusions

The purpose of this paper is to introduce and examine a set of new explicit one-step numerical schemes. We propose these methods to solve the ODE system which is obtained after spatially discretizing diffusion term. We prove that one-stage "constant-neighbour" (CN) method is first order, while the two-stage "CpC" methods are second order for the heat equation in the time step size $h$ for arbitrary parameter values $p$. For $p \geq 0.5$ the variables at the end of the time-step are the convex combinations of the values at the beginning of the time step for the heat equation in the absence of external sources; thus, the methods are stable for arbitrarily large values of the time step size. This holds for the nonlinear Fisher's equation as well, if one treats the nonlinear term as we suggest. These statements have been proved analytically and confirmed by several numerical experiments.

The schemes have been verified by comparisons to analytical solutions of the inhomogeneous heat equation and Fisher's equation. The performance of the methods has been tested in the case of two discretized two-dimensional systems with highly inhomogeneous random parameters and initial conditions. The obtained data show that if quick results are required, the proposed schemes have a significant advantage in speed, even without adaptive stepsize control, parallelization or vectorization. We must add that they are easy to implement and parallelize and they can be applied regardless of space dimensions and

grid irregularity. On the one hand, increasing number of cells makes our methods more competitive against the widely used implicit schemes. On the other hand, if the range of the eigenvalues of the system matrix is wide (which is in connection with stiffness), the new methods have a significant advantage over the traditional explicit schemes, which are only conditionally stable. However, when highly accurate results are required, first and second order methods are not the best choice; thus, we have already started to search for higher-order versions of the new methods.

**Author Contributions:** Conceptualization and methodology, E.K.; software, Á.N.; validation, E.K. and M.S.; formal analysis, E.K.; investigation, Á.N. and M.S.; resources, E.K.; data curation, Á.N.; writing—original draft preparation, E.K.; writing—review and editing, M.S.; visualization, M.S.; supervision, E.K.; project administration, Á.N. All authors have read and agreed to the published version of the manuscript.

**Funding:** The research was supported by the ÚNKP-21-3 new national excellence program of the ministry for innovation and technology from the source of the national research, development and innovation fund.

**Data Availability Statement:** Data is available at the following link: https://github.com/Drendre/New-CpC-method-dataset/tree/main (accessed on: 15 September 2021).

**Conflicts of Interest:** The authors declare no conflict of interest.

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
