# Peer review of "A Set of New Stable, Explicit, Second Order Schemes for the Non-Stationary Heat Conduction Equation"

_mathematics, doi:10.3390/math9182284_

Round 1
Reviewer 1 Report
Find the attachment.

Reviewer 2 Report
Please see my attached referee report

Author Response
Thank you for your positive attitude towards our manuscript.
“The comparisons are made with MATLAB solvers. I do not see (maybe I missed it) where the presented algorithm was implemented. Is it implemented in MATLAB too? This should be clearly stated because plays an important role in the comparisons as far as the speed of the calculations is concerned.”
The reviewer is right and we inserted a sentence into the text: “We note that all simulations are performed using the MATLAB R2019b software on a desktop computer with an Intel Core i3-8100 as CPU.”
Reviewer 3 Report
See the attached file

Author Response
Thank you for your positive attitude towards our manuscript.
We have made some changes to further improved the manuscript.
Round 2
Reviewer 1 Report
The corrections made are satisfactory, so I recommend the publication of the revised version of the paper.
However, the following corrections are needed before publication:
- The reference 16 should be dropped since it is identical to 15;
- In Theorem 1 (Line 150) - `constant' should be replaced by `nonsingular' since this assumption is used in the proof.
Author Response
We thank the reviewer for the constructive comments.
1. Reference 16 was mistakenly filled with the data of Ref. 15. Now, this error is fixed and the paper of Xue and Feng is cited.
2. We inserted the word 'nonsingular' into the place where the reviewer proposed.